# Engaging Precision Phenotyping to Scrutinize Vegetative Drought Tolerance and Recovery in Chickpea Plant Genetic Resources

**DOI:** 10.3390/plants12152866

**Published:** 2023-08-04

**Authors:** Madita Lauterberg, Henning Tschiersch, Roberto Papa, Elena Bitocchi, Kerstin Neumann

**Affiliations:** 1Leibniz Institute of Plant Genetics and Crop Plant Research (IPK), 06466 Gatersleben, Germany; lauterberg@ipk-gatersleben.de (M.L.);; 2Department of Agricultural, Food and Environmental Sciences, Università Politecnica delle Marche, 60131 Ancona, Italy

**Keywords:** chickpea, image-derived traits, growth dynamics, plant genetic resources, drought stress, chlorophyll fluorescence

## Abstract

Precise and high-throughput phenotyping (HTP) of vegetative drought tolerance in chickpea plant genetic resources (PGR) would enable improved screening for genotypes with low relative loss of biomass formation and reliable physiological performance. It could also provide a basis to further decipher the quantitative trait drought tolerance and recovery and gain a better understanding of the underlying mechanisms. In the context of climate change and novel nutritional trends, legumes and chickpea in particular are becoming increasingly important because of their high protein content and adaptation to low-input conditions. The PGR of legumes represent a valuable source of genetic diversity that can be used for breeding. However, the limited use of germplasm is partly due to a lack of available characterization data. The development of HTP systems offers a perspective for the analysis of dynamic plant traits such as abiotic stress tolerance and can support the identification of suitable genetic resources with a potential breeding value. Sixty chickpea accessions were evaluated on an HTP system under contrasting water regimes to precisely evaluate growth, physiological traits, and recovery under optimal conditions in comparison to drought stress at the vegetative stage. In addition to traits such as Estimated Biovolume (EB), Plant Height (PH), and several color-related traits over more than forty days, photosynthesis was examined by chlorophyll fluorescence measurements on relevant days prior to, during, and after drought stress. With high data quality, a wide phenotypic diversity for adaptation, tolerance, and recovery to drought was recorded in the chickpea PGR panel. In addition to a loss of EB between 72% and 82% after 21 days of drought, photosynthetic capacity decreased by 16–28%. Color-related traits can be used as indicators of different drought stress stages, as they show the progression of stress.

## 1. Introduction

In addition to the rising global population, the changing climate poses challenges for agriculture [1]. In Europe, droughts will be 11–28 times more frequent in different regions and will become much more severe in terms of their spatial and temporal spread [2]. However, there is already a high variability in the severity, timing, and intensity of droughts with severe consequences for Europe [3]. This is especially true for legumes, as they are summer crops, which are more yield-instable than winter crops [4,5,6]. Many studies among several crop species have employed plant genetic resources (PGR) to detect quantitative trait loci (QTL) that can harbor beneficial alleles at loci for relevant phenotypic traits under challenging environments [7,8,9,10]. In chickpea, valuable QTLs from landraces against abiotic and biotic stress factors and improved root growths have been introgressed by marker-assisted backcrossing into cultivars [11,12]. Other methods for the introgression of regions of interest or more specific genes include CRISPR [13,14]. However, besides drought tolerance per se, it is also essential to further investigate the ability to recover from drought, as precipitation is becoming more extreme and unpredictable [15]. Hence, it is necessary to evaluate PGR not only for drought tolerance but also for drought recovery and to analyze drought stress in the vegetative stage in addition to the drought stress at the end of the season [16]. So far, it is not sufficiently understood how plants respond to drought stress at different growth stages and during recovery to contribute to a more resilient agricultural system.

Chickpea (*Cicer arietinum* L.) is one of the legumes of the future [17,18,19]. Global production was 15 million tons, with cultivation being carried out primarily in India and Turkey, followed by Pakistan, Myanmar, and Ethiopia; an increasing production can be observed in Europe [20,21] (FAO, 2021). With their high protein content and important nutrients, they are not only in line with the dietary trend towards less meat consumption but because they form symbiotic nodulation with bacteria, they require less fertilizer and thus fit into an agriculture in which the use of fertilizers is viewed critically [22,23,24]. There are two different types of chickpea: *desi* and *kabuli*. The rough grains of *desi* are different in color and smaller than the beige and softer-coated *kabuli*. In addition, there are differences in biomass formation, in the metabolomic response to drought stress, and in anthocyanin synthesis [25,26,27]. To increase chickpea production in Europe, studying drought stress during the vegetative stage is crucial, as is the development of genotypes that are less sensitive to cold and more tolerant of *Ascochyta blight* [28,29,30,31]. The response of chickpea to drought stress depends on its duration and intensity, as well as the growth stage of the plants [18]. Terminal drought stress, which occurs from the early pod set, reduces biomass, reproductive growth, the harvest index, and final seed yield [32]. A higher abscisic acid content was also found in the seeds, which presumably leads to pod abortion. Reduced grain yield could be explained by a reduced growth rate, the leaf area index during the pod-filling stage, and reduced biomass during the reproductive growth stage [33]. Field studies have shown that chickpeas develop a deeper and denser root system to absorb water from deeper soil regions and that they tend to deposit lignin in the root xylem when exposed to drought stress [34,35,36]. Furthermore, drought stress has a negative impact on nodulation, which has a negative impact on yield [37]. Animoacids, especially asparagine, and organic acids such as malate accumulate in the nodules and lead to inhibition of respiration, nitrogen accumulation, an imbalance in the cell redox status in the nodules, and reduced nitrogenase activity [38].

Until now, little has been known about drought tolerance during the vegetative stage, because most studies have focused on terminal drought stress, which is prevailing in the major production areas and to which chickpea is adapted. In addition to the rooting ability, a low relative loss of above-ground biomass is an important criterion to assess drought tolerance. Losses in biomass production due to drought stress are primarily due to the inhibition of photosynthesis since it is the basic process for maintaining plant growth [16,32]. To evaluate photosynthesis, the pigments, chlorophyll, and carotenoids, which play a role in light trapping and photoprotection of the photosynthetic apparatus, were considered [39]. Recently, chlorophyll fluorescence traits were studied and provided insights into photosynthetic activity, especially under stress conditions [40,41,42]. To keep up with the progress of other technologies, the measurement of chlorophyll fluorescence was implemented in high-throughput phenotyping (HTP) [43].

The field of genotyping has developed tremendously in recent years with decreasing costs and increasing precision, but the acquisition of phenotype expression is often even more complex, objective, and costly [44,45]. HTP is a way to evaluate genotypes in detail, particularly under controlled greenhouse conditions [46]. Traits such as plant height, color-related traits, or most importantly, the Estimated Biovolume (EB) are assessed [47]. So far, in chickpea, HTP has only been employed to evaluate salinity tolerance and to detect genetic loci for growth rate, water use efficiency (WUE), and the number of seeds under salinity and control conditions, among others [48,49]. No study on chickpea for drought stress tolerance employing HTP has been conducted yet.

Numerous HTP studies with high data quality on cold, heat, or drought stress, for example in pea (*Pisum sativum*), *Arabidopsis thaliana*, barley (*Hordeum vulgare*), and maize (*Zea mays*), were conducted to search for tolerance traits or combined with genetic data in a genome-wide association study (GWAS) to decipher relevant QTL with spatial and temporal precision [50,51,52,53]. HTP has the strength to assess the temporal dynamics of agriculturally important traits such as biomass or plant height and reveal traits related to plant physiology such as WUE and the efficiency of photosynthesis [50,54]. Furthermore, color-related information was used to study the progress of drought stress and senescence in barley (*H. vulgare*), rice (*Oryza sativa*), maize (*Z. mays*), and wheat (*Triticum aestivum, T. durum*) [47,53,54,55].

The study is aimed at the investigation of vegetative drought tolerance and recovery in a panel of PGR of chickpea by employing HTP. The panel is balanced for the two types of chickpea, *desi* and *kabuli,* and was selected to maximize the genetic diversity of the species on the basis of passport data. The study will reveal (1) how drought during the vegetative stage affects chickpea growth performance and physiology, (2) how chickpea is able to recover from this type of drought, and (3) how to identify superior genotypes for further studies within the panel of PGR.

## 2. Results

### 2.1. Data Quality and HTP Experiment

The data quality, represented here as heritability, was high for the EB throughout the whole experiment (Figure 1). Heritability for the stress treatment was higher than that of the control treatment, averaging 0.80 compared to 0.55 in the control.

The heritability of the manually determined plant weights was in line with EB results (Appendix A). The data quality of the PH was very high with a heritability of over 0.75 throughout all days (Appendix A). Similarly, the heritability for the MCV was approximately 0.7, except for some initial days and the last days during the recovery phase in the drought stress treatment. Furthermore, the heritability for r2g was very high with over 0.75 for the control treatment, but generally lower in the drought stress treatment. In contrast, heritability for y2g was higher in the drought stress treatment, primarily towards the end. For the traits obtained from chlorophyll fluorescence imaging, the heritability was also high, with averages of 0.76 and 0.65 in control and stress treatments, respectively (Appendix A).

The EB of the last imaging day DAT 43 has been correlated with the manually measured plant weights; high coefficients of correlation (*r* = 0.96 and 0.97) were revealed for plant dry weight and fresh weight, respectively (Figure 2; Appendix A).

In conclusion, all investigated traits have sufficient data quality for almost all days and can be used to gain further insights into drought effects and trait relations.

In the first experiment, no nodulation was observed in a single plant. Since the second experiment was a repeat of the first experiment with the same conditions, it was assumed that no nodulation occurred here either.

### 2.2. Impact of Drought

When looking at the EB and the manually recorded plant weights in both treatments, it is clear that lowering the PAW had a significant influence on the formation of EB (Appendix A). Thus, a negative RGR for drought stress was observed at DAT 22 (Appendix A). This was followed by another week of very low RGR until it finally increased, with the onset of re-watering for the recovery (Appendix A).

Thus, plants in the two treatments differed significantly in EB after seven days of drought from DAT 14 onwards (Figure 3). EB was 18.5 × 10^−5^ voxels for the drought stress treatment at DAT 28, the last day of drought, and 85.1 × 10^−5^ voxels in the control treatment. Therefore, the drought period resulted in an average of 78% loss of EB (Appendix A).

The CV for EB was similar for both treatments up to DAT 12, but in the further course, the CV for the control treatment was 30% higher than that for the drought stress treatment, which was only approximately 20% (Appendix A).

Drought stress also had a significant effect on PH, MCV, r2g, and y2g (Figure 4 and Appendix A). For PH, stressed plants were significantly smaller than unstressed plants from DAT 16 onwards. At the end of the drought period, the stressed plants were 32% smaller than those in the control treatment. A significant treatment difference in MCV was observed between DAT 16 and DAT 36. Initially, the MCV in stress increased but then the loss remained relatively constant at −6.4 to −5.79% during the last days of drought stress from DAT 24–28. With the onset of re-watering, the MCV in stress decreased again until there was no significant difference to control from DAT 36 onwards. The r2g ratio was significantly higher in stress treatment from DAT 17 until the end. It was 0.005 for the control and 0.013 for the stress treatment on the last day of drought, representing a loss of −160% and indicating an increase compared to the control. As the recovery progressed, the r2g of stressed plants decreased again, so that the r2g on DAT42 was 0.004 for stressed plants and 0.006 for non-stressed plants. A very similar pattern was observed for the y2g. For y2g, on DAT 28, the control was 0.029 and the drought stress treatment was 0.069, showing a loss of −138%.

The CV for PH was relatively constant throughout the whole experiment in the control treatment but increased continuously from DAT 16 onwards in stress up to 18% at DAT 42 (Appendix A). For the MCV, the CV was higher for the drought stress treatment compared to the control treatment for DAT 12–22 and 41 and 42, but for the remaining days, from DAT 22–40, the opposite was true. Overall, the CV for MCV was the lowest of all considered traits, ranging between 1.5 and 2.5%. The CV for the r2g increased in both treatments from 13% on DAT 2 to 25% on DAT 15. For the following DATs, it was higher for the control treatment by up to 7% until, finally, for the last DAT 40–42, the CV was higher in the drought stress treatment. The CV pattern for y2g was similar to that of r2g, but continuously higher in the drought stress treatment from DAT 22 onwards.

The WUE in the three different drought phases was evaluated (Figure 5, Appendix A). The mean during DA was 0.038 voxel/mL for the control and 0.039 voxel/mL for drought stress (Appendix A). During DR, the WUE was lower in drought (0.026 voxel/mL) compared to the control treatment (0.04 voxel/mL). In DT, WUE was significantly higher in drought stress (9.63 voxels/mL) than in the control treatment (0.033 voxels/mL).

The ΦPSII imaging results showed differences between the two treatments, some of which were significant (Figure 6, Appendix A). ΦPSIIh decreased with continued drought stress and increased again with re-watering. At DAT 20 and 27, after 13 and 19 days of drought stress, respectively, the control treatment and the drought stress treatment differed significantly. The loss in the stressed plants was 7% on DAT 20 and 20% on DAT 27 of the ΦPSIIh in the control treatment (Appendix A).

For the ΦPSIIr and ΦPSIIh, comparable results and significances were found (Appendix A).

### 2.3. Correlations between Traits

Correlation coefficients *r* between EB and PH, r2g, and y2g traits over each phase of DA, DT, and DR have been calculated (Appendix A). For PH and EB, *r* ranged from 0.56 to 0.87 in all phases and both treatments. In contrast, there was a negative correlation between the MCV and EB in both treatments, for DA and DR. In DT, there was a significant difference between the treatments. For the control treatment, *r* was −0.54, but for DT in the drought stress treatment, it was 0.53. Considering the EB and r2g traits, *r* was strongly negative (approximately −0.7 for all combinations of treatment and phase); only for the DR phase in the control treatment did *r* result in −0.54. All *r* values for all the pairs of traits for each phase and treatment yielded significant results, with the only exception being *r* between EB and y2g for the DR phase and control treatment (*r* = 0.005; *p* = 0.88). In contrast, for DR in drought stress, there was a strong negative correlation between y2g and EB (*r* = −0.74; *p* < 0.0001); for the DT, *r* between y2g and EB was −0.75 for the control treatment but only −0.33 in the drought stress treatment, even if both were significant.

The coefficient of correlation *r* between ΦPSIIh and ΦPSIIl was 0.5, while *r* between ΦPSIIr and ΦPSIIh was significantly higher, equal to −0.94 (Appendix A).

Furthermore, some correlations between ΦPSIIh and image-based traits were significant with persistent drought stress (Appendix A). On DAT 27, after 19 days of drought stress, the correlations of ΦPSIIh to EB and PH were significantly positive (*r* = 0.69 and *r* = 0.55), and to the color-related traits, MCV, r2g, and y2g were significantly negative (*r* = −0.62; *r* = −0.7 and *r* = −0.68; Appendix A). In addition, on DAT 27, a strong positive correlation between the color-related traits and between EB and PH was visible as well (Appendix A).

### 2.4. PGR under Drought Stress

The set of 60 genotypes could be grouped on the basis of biological status and *desi* and *kabuli* types. In particular, the differences in *kabuli* and *desi* in terms of their tolerance to vegetative stage drought stress were of interest and were evaluated based on the drought stress period DAT 8–28 (Appendix A). In the control treatment, principal component (PC) 1 explained 75%. The EB and PH traits were approximately opposite to y2g and r2g, respectively. For PC2, which explained 15.3% in the control treatment, the genotypes are subdivided primarily for MCV. A clear separation of the two chickpea types or even a tendency to be different was not detected. PC1 explained 57% of the drought stress treatment, which was less than the control treatment. Furthermore, PC1 separates genotypes based on EB and PH versus r2g and y2g. PC2 explained 19.5% of the variation and was influenced by MCV. In the PCA for drought stress, there was a slight differentiation of *desi* and *kabuli* genotypes, with *desi* mostly present in the positive range of PC1 (high EB and PH, low r2g and y2g). There were no significant differences in biological status between cultivars, domesticated material, landraces, and breeding material (Appendix A).

To evaluate whether *desi* or *kabuli* were more tolerant to drought stress, the percentage loss caused by drought was investigated (Appendix A) There were no significant differences in losses for EB, r2g, and y2g between both types. Loss of PH was lower for *desi* from DAT 16–19, but on DAT 28, a similar loss was detected for *desi* (31%) and *kabuli* (34%) types (Appendix A). Similarly, loss for MCV was significantly different from DAT 24–31 for the two types. For DAT 28, the loss between drought stress and control treatment was lower for *desi* (−5.8%) compared to *kabuli* (−7.2%).

Considering the biological status, the loss ratio of drought stress and control treatment was calculated to verify if breeding materials, cultivars and landraces, or domesticated materials are different in terms of their drought tolerance (Appendix A). However, no significant differences were detected. In addition, the genotypes of the domesticated material group were very heterogeneous.

In addition, we investigated for each individual the deviation from the mean value of the BLUEs of the entire set of materials for both treatments in relation to their EB (Figure 7). At DAT28, the last day of stress, mostly *desi* genotypes had significantly higher mean values of both drought stress and control treatment. Thus, these genotypes behave significantly better than the *kabuli* genotypes. A similar result was obtained for the last day of the experiment (DAT 42), that is, after drought stress and recovery. *Desi* was significantly better than *kabuli* over the mean of all 60 genotypes.

In contrast, no significant differences in terms of deviation from the mean values of BLUEs for EB in both treatment conditions were detected considering the different biological statuses of the genotypes (Appendix A).

Comparing the WUE for the *desi* and *kabuli* types, a significantly higher WUE was detected for desi genotypes (11.5 voxel/mL) compared to *kabuli* ones (7.76 voxel/mL) in the drought stress treatment during the DT phase (Figure 8 and Appendix A). No significant differences in the biological status of the genotypes were detected for WUE.

When looking at the image-derived traits r2g, y2g, and MCV for the deviation from the mean values of the entire panel, the *kabuli* genotypes showed a significantly higher r2g than *desi* ones, but no significant difference was detected for MCV (Appendix A).

Concerning photosynthesis traits, the two chickpea types showed a significant difference only on DAT 34 (Appendix A): The mean ΦPSIIh was higher for *desi* (0.438) than for *kabuli* (0.425) types. When the deviation from the mean of ΦPSIIh was calculated for the *desi* and *kabuli* groups and biological status, no significant differences or trends were visible (Appendix A).

There were 32 genotypes that have better EB and WUE in the different phases of DT, DR, and DA and at DAT 28 and DAT 42 than the panel mean of 60 genotypes (Appendix A). These 32 genotypes include 10 Kabuli and 22 Desi genotypes. INCCP_00119 (*desi*, Turkey, landrace), INCCP_00139 (*desi*, Tajikistan, landrace), INCCP_00291 (*desi*, Mexico, cultivar), and INCCP_01917 (*kabuli*, Portugal, landrace) were better for WUE and EB than the mean of the panel for both treatments on the significant days DAT 28 and 42 and in all phases, DA, DT, and DR.

## 3. Discussion

HTP is considered an important tool that allows the rapid and precise testing of genotype environmental interactions [56]. In the present study, the effects of drought stress during the vegetative growth period under well-watered and drought stress conditions using non-destructive HTP was investigated for the first time in chickpea using a diverse panel of PGR. The high data quality results of the daily phenotyping showed the varying tolerance of PGR and allowed us to draw conclusions about the suitability of PGR for pre-breeding.

### 3.1. Suitability of HTP

The HTP system used in this study has already proven useful for studies to test the biomass development and color change of wheat Near Isogenic Lines throughout the life cycle in phases of varying drought stress and also the biomass development of barley PGR under drought stress in the vegetative stage [47,55]. The greenhouse conditions met the temperature requirements for chickpea and were combined with an irrigation system that simulates drought stress followed by recovery during vegetative development.

The heritability of EB in the drought stress and control treatment was comparable to results from a chickpea HTP experiment for salinity tolerance [48]. Furthermore, the heritability was also high for PH, MCV, and r2g in the control treatment. On some days, especially at the beginning and towards the end of the experiment, the heritability for the y2g, MCV, and r2g was lower in the drought stress treatment. This could be partly attributed to the non-uniform development and maturation of the PGR and the associated physiological changes in pigment composition during these days. In addition, the heritability for the chlorophyll fluorescence imaging traits was satisfactory. In general, heritability was comparable to that of previous experiments in barley on this HTP system and is suitable for future genome-wide association studies [52,55]. The high correlation of EB to measured plant weight demonstrates the suitability of EB as a proxy for biomass. Based on the high-quality dataset, statistical analyses could be carried out to evaluate the PGR in terms of their tolerance to drought stress and ability to recover after drought.

The CV was used to determine the degree of phenotypic variation over time [53]. The change in the CV reflected the variation in PGR, confirming the method and utility of this dataset for further analysis. Thus, the CV for this PGR under drought stress was highest for the r2g and y2g color ratios and also high for EB, resultant WUE, PH, and ΦPSIIh.

### 3.2. General Effect of Drought Stress during HTP Experiments

Drought stress significantly impairs plant development for several traits and yield components, making the breeding of tolerant varieties a complex task. This study highlights the importance of HTP for the screening of vegetative drought stress tolerance to identify superior genotypes within the PGR panel.

According to the timeline of the irrigation regime, statements can be made about the relevance of traits at certain DATs, e.g., at the maximum sustained drought stress on DAT 28, or in certain phases such as the DT, DR, and DA [53]. The difference between treatments became significant for each trait within a few days, but the difference in EB occurred first. EB showed a significant difference on DAT 14, i.e., 7 days after the onset of drought stress, 2–3 days before the other traits. This is slightly later compared to results in barley PGR [55]. As there is a correlation between biomass and seed yield in chickpea, the reduced EB formation in drought and low biomass of genotypes would likely result in lower seed yield [48,49].

Drought stress altered the image-based traits that differed even after re-watering. MCV was the only trait of the image-based traits for which the values of the two treatments converged again in the recovery phase, so from DAT38 onwards, there was no longer any significant difference between the treatments. To describe differences between genotypes based on percentage losses and coefficients of variation, r2g, EB, and y2g were most informative [53,54].

The effect of drought stress and the optimal choice of the timing of re-watering was shown in the RGR. As RGR became negative with continued drought stress, the objective of the study to evaluate the drought tolerance and recovery of chickpea was realized with the scheduling of irrigation. If irrigation had been delayed, some genotypes would not have shown recovery.

WUE based on EB and water use was another informative trait. It is considered for breeding plants more tolerant to drought and involves optimizing biomass accumulation and transpiration [57]. It is advisable to supplement the WUE with data on transpiration, stomatal density conductivity, or vapor pressure deficit [58]. In a field study with one chickpea variety, optimum WUE was achieved with early sowing and increased irrigation [59]. Furthermore, in chickpea, better adaptation to water deficit was associated with higher relative water content, longer chlorophyll retention, and higher osmotic adjustment [60]. The significant differences and variations in WUE during DT can be used to select suitable PGR genotypes.

PH is an important trait that correlates with shoot biomass and thus also with seed yield [49]. Drought reduced PH in our study, and the PGR panel showed substantial phenotypic variation in PH and losses in PH.

Color-related traits have been used previously to represent physiological responses to challenging environmental conditions in HTP in wheat (*T. aestivum*; *T. durum*), barley (*H. vulgare*), and maize (*Z. mays*), as well as rice (*O. sativa*) and lettuce (*Lactuca sativa*) [47,54,55,61,62].

Under drought, both color ratios, y2g and r2g, increased, indicating pigment changes such as chlorophyll, carotenoids, and anthocyanins, which play a role in plants’ reactions to stress and starting senescence symptoms [63]. Carotenoids, for example, stabilize the lipid membrane, are important for photosynthetic light collection, and protect photosystems from oxidative damage caused by light radiation [63]. Similarly, anthocyanins, which are of red to blue color, reduce the photoinhibition and photobleaching of chlorophyll and occur in response to environmental extremes [64,65]. In addition to anthocyanin, in chickpea, lower chlorophyll and carotenoid content was observed under drought stress conditions in the field, which could explain the changes in r2g and y2g [66,67].

The MCV first increased under drought stress, indicating a deeper green color of the leaves, then remained constant from DAT 24–28 during advanced drought stress, and finally fell back to the value of the control treatment with the resumption of normal watering. In principle, there is a high correlation between hue value and chlorophyll concentration, which has been observed in a wide range of species, including tobacco (*Nicotiana*), grapevine (*Vitis labrusca*), or *Arabidopsis thaliana* [65,68,69]. The initially darker green shade of the leaves could be explained by the fact that under drought stress, the water content in the cells decreased and therefore the chlorophyll content increased in relative terms [67]. Another reason could be the short-term overcompensation of chlorophyll, which has already been observed in soybean (*Glycin max*) under drought stress [70]. The constant hue value of DAT 24–28 could be due to anthocyanin accumulation, which correlates negatively with hue [65].

Many studies in greenhouse and field conditions demonstrated a negative effect of drought stress on photosynthesis, e.g., in soybean (*G. max*), lettuce (*L. sativa*), and wheat (*T. durum*) [16,62,71]. In this study, the lower ΦPSIIh, which represents a lower light quantum yield, showed a reduced photosynthetic capacity. However, in our study, when plants recovered from drought, photosynthetic capacity had not been permanently damaged and ΦPSIIh returned to well-watered levels, similar to a field study with soybean (*G. max*) and drought stress [16]. Similarly, [47] ΦPSIIr was higher under drought stress.

In line with the study for salinity tolerance in chickpea, EB and PH were always positively correlated in control and drought stress [49]. While correlations between EB and the other imaging traits were quite constant, correlation to MCV showed a different pattern. The delayed change from a positive to a negative correlation with the onset of re-watering suggested that MCV might be important for the selection of PGR for drought stress tolerance and was found for the DT phase. Y2g and r2g showed a more durable correlation with EB and y2g appeared to be more informative for DR, with a defined shift in correlation between control and drought stress. The color traits had a high heritability, which is in accordance with studies in barley and maize and therefore could be used in evaluating the wilting process [55,72].

Precision phenotyping thus allows one to select the most suitable traits, but moreover, also determines the critical moment for their evaluation [73].

In addition, the use of chlorophyll fluorescence imaging enabled a complex analysis of chickpeas under drought stress and highlights the importance of studying with complementary methods [43,50,54,62]. A high correlation between plant area and chlorophyll fluorescence imaging has been noted previously, which is comparable to the EB and PH traits, which have a high correlation to ΦPSIIh [54].

### 3.3. PGR for Drought Stress Tolerance during Vegetative Phase

The diverse PGR panel showed high phenotypic diversity throughout the whole experiment. Grouping according to geographical area and biological status revealed no differences in performance under well-watered or drought stress conditions, which could be due to the sample size.

However, significant differences are known for the two types of chickpea, *desi* and *kabuli*. *Desi* has been reported to be more tolerant to drought stress than *kabuli* [25,74]. A higher dry weight in the seedling stage, specific leaf area, and reduced growth for the *desi* type were observed in pot and climate-chamber experiments [25,74]. In our panel, no significant difference was found between *desi* and *kabuli* in the image-derived traits for the entire course of the experiments. When evaluating the performance in comparison to the deviation from the panel mean values on specific DAT, *desi* genotypes behaved significantly better for EB on DAT 28 and 42 and WUE during DT. This was in agreement with earlier studies that identified a higher WUE and transpiration efficiency in *desi* and could be in part attributed to the anatomy of the xylem vessels and cortical layers of roots, which have less resistance to water [25,74].

In general, the differences between the two types, *desi* and *kabuli*, have been intensively discussed in the literature. Re-sequencing of 29 varieties revealed that only 2% of the genomes are different regions and these are likely signatures of selection during improvement [75]. Interestingly, markers for proanthocyanin were found to be significantly different for *desi* and *kabuli,* with *kabuli* showing a reduced function for blocking transcription factors for anthocyanin biosynthesis [26]. In this study, *kabuli* showed a significantly higher deviation from the mean of the panel for the r2g value than *desi* at DAT 28 under drought stress. This higher r2g value could be due to a higher anthocyanin content. However, there was no significant difference for MCV at DAT 28 under drought stress compared to the mean of the panel between *desi* and *kabuli*, although the hue value is correlated to anthocyanin content [65].

In contrast to [74], our study did not reveal a significantly better photosynthetic performance of *desi* types under drought stress. This could be due to the different genotypes, type of stress timing, and severity or measurement of photosynthesis. *Desi* tended to perform better photosynthetically during the late drought stress phase and during recovery in the context of our study.

Based on a pre-screening, we were able to gain insights into the suitability of PGR for drought stress and recovery. For the relevant trait EB, there were 22 *desi* and 10 *kabuli* genotypes that were better than the mean of the panel at key DAT 28 and 42 and for the different phases DR, DA, and DT. Four superior genotypes are identified that can be used for future improvement of vegetative drought tolerance. The four genotypes INCCP_00119 (*desi*, Turkey, landrace), INCCP_00139 (*desi*, Tajikistan, landrace), INCCP_00291 (*desi*, Mexico, cultivar), and INCCP_01917 (*kabuli*, Portugal, landrace) showed superior WUE and EB and are valuable genotypes for further studies.

## 4. Material and Methods

### 4.1. Plant Material

The set of materials used in the present study consisted of 60 chickpea accessions (Appendix A). Each accession is represented by single seed descent (SSD) material derived by at least two cycles of selfing from 12 accessions from the IPK Gatersleben genebank and 48 accessions from the Western Regional Plant Introduction Station (USDA-ARS, Washington State University, Pullman, WA, USA) [20,21]. These materials represent a subset of a larger collection (Training CORE, T-CORE) developed within the INCREASE (Intelligent Collection of Food Legumes Genetic Resources for European Agrofood Systems) [21,76] and EMCAP (European and Mediterranean Chickpea Association Panel) projects [20]; and is being tested in field experiments in collaboration with partners in Italy, too. The 60 genotypes were selected to maximize the genetic diversity of the T-CORE using passport data; in particular, the genotypes originated from 39 countries and 16 regions worldwide. Moreover, the set of materials is balanced for being *desi* (30 genotypes) and *kabuli* (30 genotypes). Considering the biological status, the set is composed of 3 breeding materials, 13 cultivars, and 42 landraces; for the remaining two genotypes, the biological status is domesticated material (Table S12).

### 4.2. HTP Experiments

The HTP system (LemnaTec-Scanalyzer 3D) used in the present study is installed in an environmentally controlled greenhouse at IPK Gatersleben (51°49′23″ N, 11°17′13″ E, altitude 112 m). In this system, each plant was transported by conveyor belts to the imaging chambers equipped with top and side view RGB (Red, Green, Blue) and fluorescence cameras, where a lifter allows imaging from different angles in side view. The balance-watering station enables controlled watering and thereby defined drought setups.

Plant material was tested in two experiments with two biological replicates per genotype and treatment. The first experiment was conducted from 24 March 2021 to 19 May 2021 and the second from 10 June 2021 to 5 August 2021. Two experiments were planned to obtain a total of four biological replicates per genotype and treatment. For both experiments, two seeds were sown directly into the pots and thinned out to one seedling per pot after emergence. Each pot (18.5 cm height × 14.9 cm diameter) was filled with Klasman substrate No. 2 described in [47]. After 14 days of pre-cultivation in a regular greenhouse chamber outside the HTP system at 24 °C during the day and 20 °C during the night, with a relative humidity of 67% during the day and 76% during the night, a daylight period of greenhouse lights of 15 h (from 6 am to 9 pm), and manual watering, the plants were transferred to the greenhouse with the HTP system with the same growing conditions. To each pot, 7 g of fertilizer with a composition of 19% total nitrogen, 9% P_2_O_5_, and 10% K_2_O was added, and no inoculation was carried out to promote nodulation. Nevertheless, each plant in the first experiment was examined for nodulation after the experiment was completed. A plant support was placed on each pot and each pot was placed into a tray so that any water added could be absorbed by the plant. During the experiment, LemnaTec software was used to randomize the arrangement of the plants twice a week resulting in a fully randomized design. After an establishment phase to bring all plants to the same level of plant-available water (PAW) of 70%, the irrigation level was lowered to 10% from day 8 after transferring (DAT) for plants in the drought stress treatment (Table 1; SM S2). The watering regime and simulation of drought stress were developed on this HTP system and have already been published [52]. The plants of the control treatment were maintained at 70% PAW from DAT 1 until the end at DAT 42 (SM S2).

On DAT 29, gradual re-watering was planned for both experiments with 300 mL initially, followed by irrigation to 70% on DAT 30. The irrigation was performed in two steps to allow the plants to slowly absorb the water. Information on daily watering based on weight before and after watering can be extracted with the system software. At the experiment’s end, fresh and dry weights of the above-ground shoot part were determined. Furthermore, due to technical difficulties, only an incomplete set of images could be recorded on DAT 1 for experiment 1 and on DAT 14 for experiment 2. These two days were excluded from the analysis.

### 4.3. Image-Derived Plant Traits

The images were analyzed using the IAP version 2.3.0 (Integrated Analysis Platform (IAP)) [77]. The traits used here include Estimated Biovolume (EB, [voxel 10^−5^]), Plant Height (PH; [mm]), Mean Color Value (MCV; [hue]), the red to green ratio (r2g; [%]), and the yellow to green ratio (y2g; [%]) [55,77]. The MCV refers to the HSV color space. A 20-bin histogram was calculated, which provides information about the composition of the detected plant color [77]. Based on this model, a mean hue of 0.23 corresponds to an image of a green plant. The values y2g and r2g indicate the percentage of yellow and red pixels in the image, respectively. The EB was calculated from the images from the top view camera and the images of three side views:$$Estimated\;Biovolume\;\left[voxel\right]=\sqrt{average\;pixel\;side\;area^{2}*top\;area}$$

The PH results from pictures of the side view and the y2g, r2g and MCV result from pictures of the top view.

To determine the PAW, the pot weight before watering was taken into account, as was described in [52].

To draw conclusions about the tolerance of, for example, the two chickpea types or biological status, the loss was calculated as the ratio of drought stress to control for the imaging traits.
$$Loss\;of\;Trait\;\left[\%\right]=\left(1-\frac{trait_{drought\;stress}}{trait_{control}}\right)\times 100$$

Based on the mean of the BLUE values in the two experiments for the 60 genotypes, the relative growth rate (RGR) was calculated. The missing values at DAT1 and 14 were linearly interpolated to calculate the RGR.
$$Relative\;Growth\;Rate\;\left[\;\frac{voxel}{DAT}\right]=\frac{\ln\left(EB_{i}\right)-\ln\left(EB_{i-1}\right)}{DAT_{i}-DAT_{i-1}}$$

For further analysis, the DATs were divided into phases. DAT 2 to 7 refers to the establishment phase, DAT 8–28 to the drought tolerance phase (DT), DAT 29–42 to the drought recovery phase (DR), and the period from DAT 8–42 was considered the drought adaptability phase (DA).

The WUE was calculated as EB per milliliter of water during each of the phases of DA, DT, and DR. If the irrigation was at zero milliliters, the amount of irrigation for the entire phase has been set to 1 milliliter to proceed with further analysis. Outliers which were detected for EB were removed from the irrigation data, and the mean for each genotype and each DAT was calculated. For each drought phase, the difference in the BLUEs of EB between the first and last DAT was calculated and divided by the sum of the irrigation to calculate the WUE.

### 4.4. Chlorophyll Fluorescence Imaging and Image Analysis

The system was supplemented with a chlorophyll fluorescence camera (FluorCam; version 7) from Photon Systems Instruments (PSI; Brno, Czech Republic) to measure photosynthetic performance from the top view. The FluorCam data were analyzed using the manufacturer’s software Plant Data Analyzer (version 3).

These measurements took place at DAT 6 (2 days before drought stress); DAT 13 (five days of drought stress); DAT 20 (twelve days of drought stress); DAT 27 (19 days of stress); and DAT 34 (five days after re-watering) (Table 1). During these days, normal imaging was advanced to 00:01 am instead of 7 am to allow FluorCam measurements at 8 am (duration 12 h).

Chlorophyll fluorescence measurements were taken approximately once per week using FluorCam similar to [43]. For the measurement, the plants were first adapted to a light intensity of 800 µmol/m^2^/s in an adaptation tunnel. This light intensity is higher than that in the greenhouse (during the growth). After the adaptation of five minutes, the plants were moved to the measuring chamber and illuminated once more for 10 s with a light intensity of 800 µmol/m^2^/s. At the end of this phase, a first saturating light flash of 4000 µmol/m^2^/s was applied to measure the operating efficiency under high light intensity (ΦPSIIh; µmol/m^2^/s). This was followed by ten seconds of 80 µmol/m^2^/s to allow the plants to adapt to low light conditions, and then a second light flash of 4000 µmol/m^2^/s was used to measure the operating efficiency under low light conditions (ΦPSIIl; µmol/m^2^/s).

The ΦPSIIl to ΦPSIIh ratio was calculated in order to measure the plasticity of photosystem II to fluctuating light (ΦPSIIr).

### 4.5. Statistics

The EB was divided by a factor of 10^−5^. For further statistical analysis, R studio version 4.1.2 was used with the packages “tidyr”, “dplyr”, “stringr”, “data.table”, “multtest”, “agricolae”, and “lattice”. Figures were created with the packages “ggplot2”, “ggpubr”, and “ggrepel”. The package “ASReml” was used to calculate the outlier, the heritability, and the best linear unbiased estimators (BLUEs). Across the two experiments, outliers have been detected separately for each treatment. In the mixed model, the genotypes are fixed effects and the experiment and the genotype and experiment interaction were random factors. For every DAT, outliers were calculated separately.

To calculate the broad-sense heritability *H*^2^, each DAT is a fixed effect, and the experiment, the genotype and experiment, and genotype interactions have been taken as random effects.

*V_G_*, *V_e_,* and *V_GxE_* are the variance components of the genotype, the genotype × experiment interaction, and the residual, respectively. *numExp* is the number of experiments for the respective DAT, and *numRep* is the number of biological replicates.
$$H^{2}=\frac{V_{G}}{V_{G}+\left(\frac{V_{G\times E}\;}{numExp}\right)+\left(\frac{V_{e}}{\left(numRep\times numExp\right)}\;\right)}$$

To calculate the BLUEs across the two experiments from the cleaned dataset, genotypes were used as fixed factors, similar to the outliers, and the experiment and the genotype–experiment interaction are random factors. The BLUEs across the experiments were used for all further analyses.

Pearson correlations with the coefficient of correlation *r* were estimated. The coefficient of variation (CV) has been calculated by the ratio of σ to µ. For the chlorophyll fluorescence values, an ANOVA followed by Tukey’s test was performed to calculate the significance levels. Significant differences were highlighted by calculating the 95% confidence intervals using the package “Rmisc”. The confidence intervals are shown as shadows in the figures. Principal component analysis (PCA) was performed using the package “factoextra”.

## 5. Conclusions

Considering the differences observed between genotypes with respect to drought stress during the vegetative stage and recovery, HTP proved to be a useful method to study these complex quantitative traits in chickpea PGR. Genotypes with superior drought tolerance could be identified from the traits of growth performance and physiology derived from the images, suggesting that further studies are needed to elucidate the underlying processes.

For practical crop improvement, this method is valuable. PGR can be assessed in an HTP experiment; by linking genotypic data, root-related data, yield-related data, and, for example, metabolomic data, QTLs can be mapped by genome-wide association studies to unravel the underlying genetic architecture for drought tolerance and recovery in chickpea. In addition, these results can then be verified through field studies.

Once potentially valuable genotypes are identified, they can be incorporated into breeding programs and introduced into elite material, for example, via CRIPSR/Cas, backcrossing with marker-assisted selection, or other breeding methods, to breed chickpea varieties that are more tolerant to drought stress in the vegetative stage and recover quickly.

## Figures and Tables

**Figure 1 plants-12-02866-f001:**
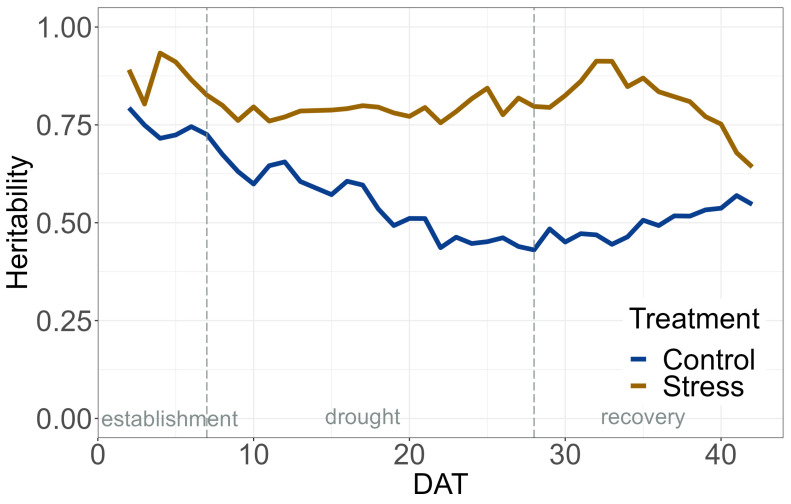
Heritability of Estimated Biovolume (EB).

**Figure 2 plants-12-02866-f002:**
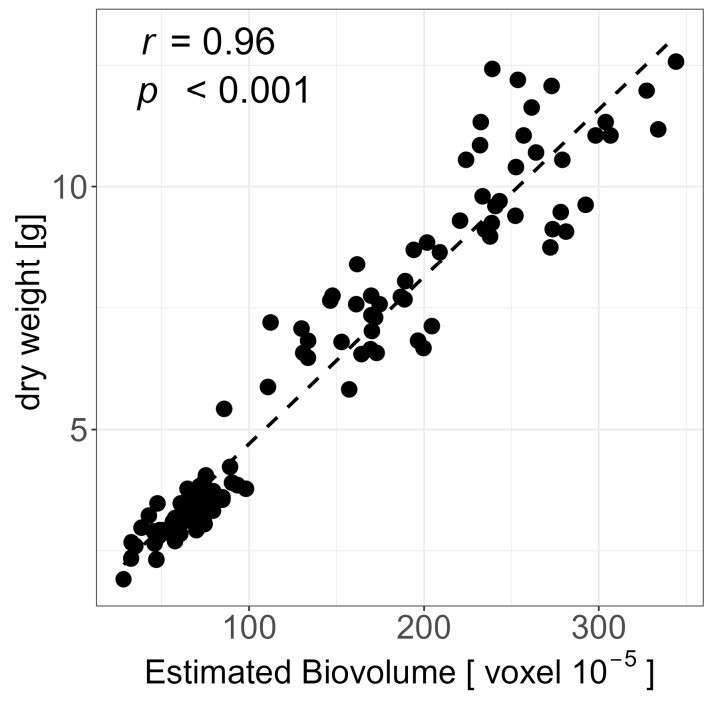
Correlation of Estimated Biovolume (EB) from DAT42 and dry weight from DAT43. Based on BLUEs across both experiments of all 60 genotypes. *p* indicates the level of significance and *r* is the coefficient of correlation.

**Figure 3 plants-12-02866-f003:**
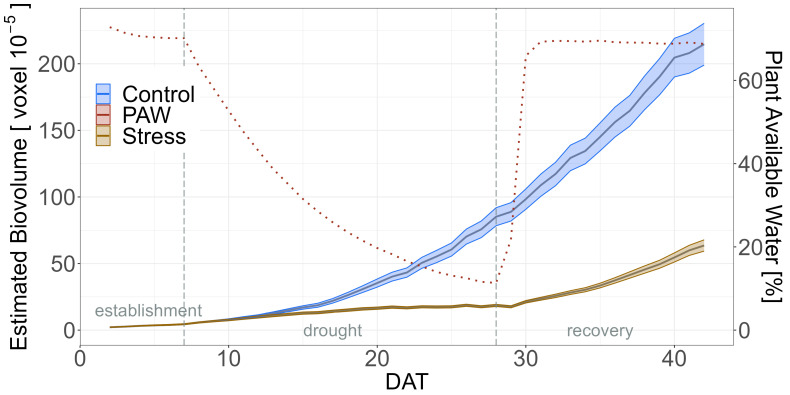
Estimated Biovolume (EB) in drought stress and control treatments. The dotted line indicates the plant available water (PAW) to which the secondary axis refers to The shadows describe the 95% confidence interval; as long as the shadows of the individual lines do not overlap, the significance level of *α* = 0.05 was reached and therefore a significant difference exists. Based on means of BLUEs across both experiments of all 60 genotypes.

**Figure 4 plants-12-02866-f004:**
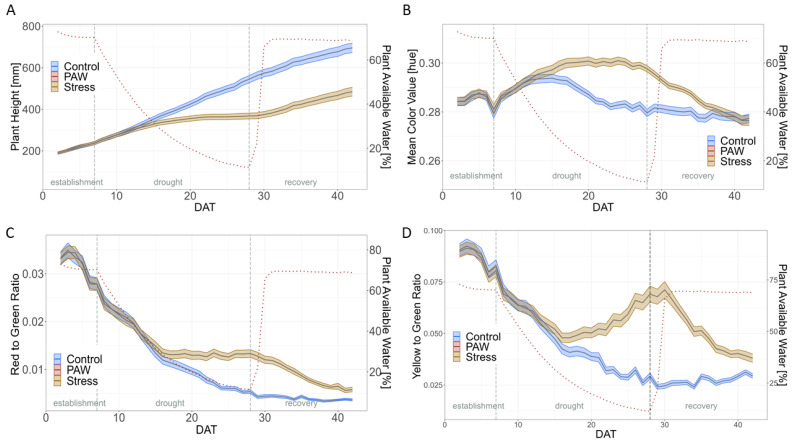
Further image-based traits under drought stress and control treatments. (**A**) Plant Height (PH); (**B**) Mean Color Value (MCV); (**C**) red to green ratio (r2g); (**D**) yellow to green ratio (y2g). The dotted line indicates the plant available water (PAW) to which the secondary axis refers to. The shadows describe the 95% confidence interval; as long as the shadows of the individual lines do not overlap, the significance level of *α* = 0.05 was reached and therefore a significant difference exists. Based on means of BLUEs across both experiments of all 60 genotypes.

**Figure 5 plants-12-02866-f005:**
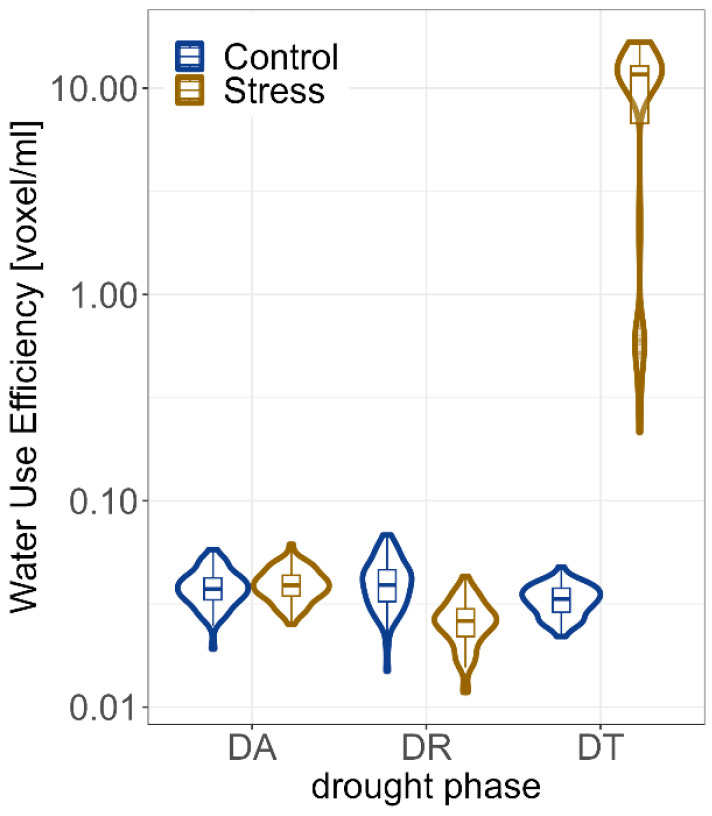
Water Use Efficiency (WUE) in different drought phases for drought stress and control treatments. Estimated Biovolume (EB) was based on BLUEs across both experiments of all 60 genotypes. The shape around the boxplot is a violin plot and describes the continuous distribution of the data at different values. DA = drought adaptability DAT 8–42; DR = drought recovery DAT 29–42; DT = drought tolerance DAT 8–28.

**Figure 6 plants-12-02866-f006:**
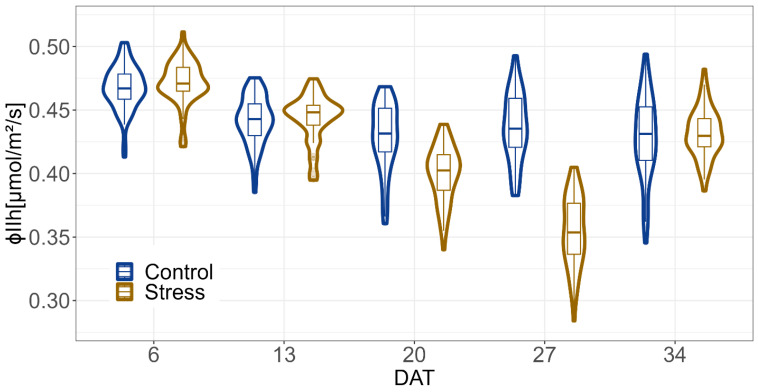
ϕPSIIh for drought stress and control treatments on several DATs. Based on BLUEs across both experiments of all 60 genotypes. The shape around the boxplot is a violin plot and describes the continuous distribution of the data at different values; 1 day before drought = DAT 6, 7 days of drought = DAT13, 13 days of drought = DAT20, 19 days of drought = DAT 27, 6 days of recovery = DAT34. ϕPSIIh = operating efficiency of photosystem II under high-light conditions.

**Figure 7 plants-12-02866-f007:**
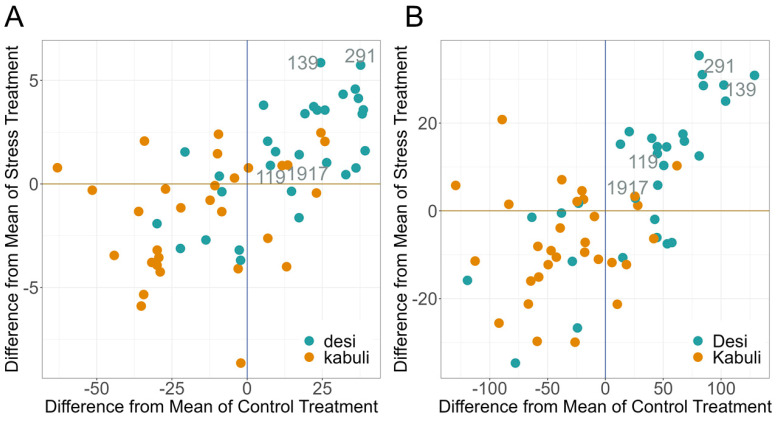
Deviation of the individual genotypes with their affiliation to *desi* and *kabuli*, from the mean value of Estimated Biovolume (EB) of all 60 genotypes. The superior genotypes were labeled with the INCCP plant material number and the labels touch the designated places. Based on BLUEs across both experiments and all 60 genotypes. (**A**) DAT 28; (**B**) DAT42.

**Figure 8 plants-12-02866-f008:**
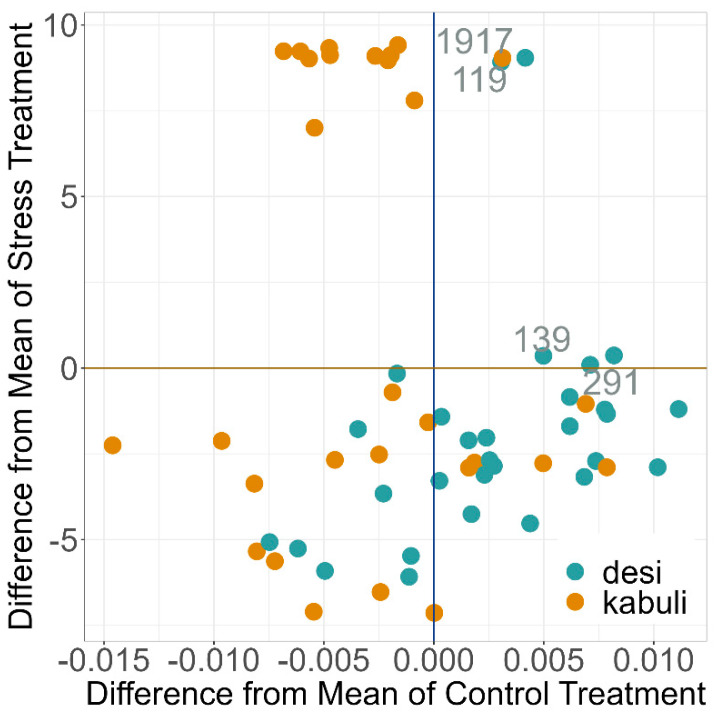
Deviation of the individual genotypes with their affiliation to *desi* and *kabuli*, from the mean value during DT of Water Use Efficiency (WUE) of all 60 genotypes. Estimated Biovolume (EB) based on BLUEs across both experiments and all 60 genotypes. The superior genotypes were labeled with the INCCP plant material number and the labels touch the designated places. DT = drought tolerance DAT 8–28.

**Table 1 plants-12-02866-t001:** Timeline of both experiments. DAS = Days after sowing; DAT = Days after transferring; PAW = Plant-available water.

DAS	DAT	Action
0		Sowing and pre-cultivation in greenhouse
14	0	Transferring to HTP system and set watering to 70% PAW
15	1	First image
20	6	Chlorophyll fluorescence measurement
22	8	Initiation drought stress: 10% PAW
27	13	Chlorophyll fluorescence measurement
34	20	Chlorophyll fluorescence measurement
41	27	Chlorophyll fluorescence measurement
43	29	First step of recovery: + 300 mL
44	30	Second step of recovery: 70% PAW
48	34	Chlorophyll fluorescence measurement
55	41	Last imaging on HTP system
56	42	Harvest

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
