# Peer review of "Engaging Precision Phenotyping to Scrutinize Vegetative Drought Tolerance and Recovery in Chickpea Plant Genetic Resources"

_plants, 2023, doi:10.3390/plants12152866_

Round 1
Reviewer 1 Report
Although the purpose of this study is to assist breeding of chickpea for drought-prone environments, I doubt that the results would be of much value to chickpea breeders for the following main reasons:
· The results only apply to a unique aerial environment, without it being related to an actual chickpea growing environment. No mention is made of relative humidity, which would be a major determinant of plant water demand. To what extent would relative genotypic differences found in this controlled environment, be repeatable in other environments, particularly in the field? Follow-up field studies would be necessary to evaluate the relevance of these CE results.
· No mention is made of roots. Root depth and proliferation pattern is a major determinant of chickpea response to terminal drought stress, and presumably also for pre-anthesis stress. To what extent does rooting pattern relate to, or perhaps override, response of photosynthesis to drought stress?
· Field grown chickpea is usually nodulated. Non-nodulated and nodulated plants have different energy and nitrogen metabolism patterns, which would affect their drought response.
· Discussion emphasizes the differences in drought response between desi and kabuli types, which has for long been well known, and related to the different environments from which those types evolved. Minimal emphasis is given to identifying genotypes with superior drought tolerance or recovery ability. Such genotypes should be identified in Figs 7 and 8.
· Figures 3 and 4 indicate relatively little difference among test genotypes in drought response, suggesting limited scope in breeding for drought resistance. Thus, discussion is needed as to whether identification of QTLs for recommendation to breeders is likely to be worthwhile, and if so what procedure should be followed.
· Use of QTLs in chickpea breeding has contributed little to chickpea yield improvement over the last three decades. Multilocation field evaluation in drought prone environments is a better prospect as it combines drought resistance and avoidance traits with other locally desired traits.
These results would only seem publishable if framed in a more holistic manner, considering the points above.
Further comments are as follows:
Materials and Methods
L144-6. What are the treatments, presumably drought stress and non-stressed control? And how did the two experiments differ? Was the second experiment a repeat of the first?
L150. Relative humidity needs to be mentioned as this determines water use. Also, the 20 deg C minimum temperature seems high for most chickpea growing environments. What actual chickpea growing environment was the controlled environment facility relating to?
L160,163. “Error! Reference source not found.”? And further throughout the text. Is this supposed to be indicating tables and figures?
L197. The relevance of colour measurements needs to be further explained here.
L219. The symbols used should be defined.
Results
L253. The relation between heritability and data quality needs further explanation, requiring elaboration of the equation at L219?
Figs 1-8 not referred to in text.
Figs 5 and 6. Need to explain the meaning of the shapes around each data point.
L283 and later. What is SM?
L351. Fig. 1 repeated, should it be Fig. 6?
Figs 7 and 8. Indicate which points are identified as drought tolerant.
Discussion
Concentrates on evaluating use of HTP for chickpea and description of the overall (across genotypes) response to drought stress and recovery. However, very little space is given to discussion of genotypic differences, L611-617 only, which is surely the main objective of the study.
References
Reference Nos. 14 and 15 are the same.
Only minor edits needed.
Author Response
My answers are written in red. The attachted file is the revised version of the manuscript. The linguistic improvements are highlighted in turquoise, required changes in pink and improvements, which I noticed in yellow.
Although the purpose of this study is to assist breeding of chickpea for drought-prone environments, I doubt that the results would be of much value to chickpea breeders for the following main reasons:
- The results only apply to a unique aerial environment, without it being related to an actual chickpea growing environment. No mention is made of relative humidity, which would be a major determinant of plant water demand. To what extent would relative genotypic differences found in this controlled environment, be repeatable in other environments, particularly in the field? Follow-up field studies would be necessary to evaluate the relevance of these CE results.
Thank you for this suggestion, the reasoning is comprehensible. We added the relative humidity in line 144. Since chickpeas are grown in the field, one approach to confirm relevance is to correlate the yield parameters of the field trial with those of the greenhouse trial. As the biomass of the whole plant was determined directly after the completion of the HTP experiment in order to correlate the estimated biovolume with the actual manually measured fresh and dried biomass (see Figure 2), it was not possible to determine the yield parameters.
The two experiments described here are part of a larger study, and the plants from the follow-up experiments were brought into a greenhouse to mature. In addition, field experiments with the same genotypes are being conducted by our collaboration partners in Italy. As soon as we have the results of all experiments, we will investigate the yield parameters of all experiments.
- No mention is made of roots. Root depth and proliferation pattern is a major determinant of chickpea response to terminal drought stress, and presumably also for pre-anthesis stress. To what extent does rooting pattern relate to, or perhaps override, response of photosynthesis to drought stress?
We have mentioned the root in passing (line 582) and have now gone into more detail (line 42 and 71-78). As it is much more difficult to study the roots, research has focused on the shoot. Thanks to new technologies and innovations, more and more studies on root development are being published. Here at IPK, we have a high-throughput phenotyping rhizotron system in the Phenosphere, where plants are studied for several weeks. Since we have promising results from the shoot HTP experiments, we plan to test the effect on root development.
However, since there can be no differences in the phenotype due to deep roots in the pot, the focus of the experiments is clearly on the shoot and the root development factor is kept more constant here than in the field.
If root development is being studied, it is certainly scientifically relevant to study photosynthesis, especially in legumes with nodulation.
- Field grown chickpea is usually nodulated. Non-nodulated and nodulated plants have different energy and nitrogen metabolism patterns, which would affect their drought response.
Thank you for these comments. Yes, nodulation is an important factor for chickpeas and was described in the manuscript in lines 74 and 77.
- Discussion emphasizes the differences in drought response between desi and kabuli types, which has for long been well known, and related to the different environments from which those types evolved. Minimal emphasis is given to identifying genotypes with superior drought tolerance or recovery ability. Such genotypes should be identified in Figs 7 and 8.
Thank you for this comment. The fact that we also find differences between desi and kabuli underlines the relevance of the HTP pot experiments. The genotypic differences according to their groupings were discussed in lines 379-444/chapter 3.4, line 438-444, 569-572 and 603-607. Additional information can be found in the supplementary material (SM 29 and 30 and SM 23, 25, 26). We have labelled the superior genotypes in figures 7 and 8 (line 414 and 427).
- Figures 3 and 4 indicate relatively little difference among test genotypes in drought response, suggesting limited scope in breeding for drought resistance. Thus, discussion is needed as to whether identification of QTLs for recommendation to breeders is likely to be worthwhile, and if so what procedure should be followed.
Many thanks for these suggestions. Figures 3 and 4 show differences between the genotypes, which are also confirmed by the coefficient of variation in the supplementary material (SM11) (line 294-296, 319-327). Moreover, the minimum loss of estimated biovolume ((1-(stress/control))*100) was 72 % and the maximum loss 82 % at DAT 28, after the longest drought stress period, as mentioned in line 26 and 287 and described in the supplementary material (SM 7).
Drought tolerance is a quantitative trait, which means that identifying QTLs is worthwhile. In line 42-45, we refer to two studies where the introduction of QLTs through marker-assisted backcrossing resulted in better performance of chickpea under stress and mention other methods.
- Use of QTLs in chickpea breeding has contributed little to chickpea yield improvement over the last three decades. Multilocation field evaluation in drought prone environments is a better prospect as it combines drought resistance and avoidance traits with other locally desired traits.
Thank you for telling from your experience. Variation is necessary to improve a crop. If we explore plant genetic resources, we will achieve more yield stability in crops. There are positive examples of integrating beneficial QTLs (line 42-45). There is not only one way to improve crops, the aim is to use and consider several complementary methods. As mentioned above under question 1, we plan to connect our results from the HTP pot experiment with those of field experiments.
These results would only seem publishable if framed in a more holistic manner, considering the points above.
Further comments are as follows:
Materials and Methods
- L144-6. What are the treatments, presumably drought stress and non-stressed control?
Thank you for your question. In the control treatment, the plants were kept at a well-watered condition. In drought stress, plants were irrigated for three weeks only when they fall below a plant-available water of 10 % (line 152-154) (SM 2).
The experimental design to simulate drought stress was developed for barley (Dhanagond et al. 2019) and is in this study presented for chickpea taking into account the relative growth rate and loss of estimated biovolume.
A drought period can be followed by precipitation during plant growth. In order to investigate the adaptability of genotypes to this phenomenon, drought stress was followed by rewatering. In order to be able to recover from a rewatering event, the drought stress should not lead to a uniform decline of all genotypes. Therefore, we chose a time for rewatering when the plants would not show any positive relative growth, but would not be exposed to the drought stress for too long.
The relative growth rate was recorded (SM 9, line 188, 281-283 and 500-504) and thus it was found that after 19 days of drought stress, the plants lost the estimated biovolume and also stopped growing slowly. Drought stress is thus definitely simulated and recovery can take place. In further studies, it would certainly be useful to simulate a stronger or longer-lasting drought stress in order to be able to evaluate additional data.
- And how did the two experiments differ? Was the second experiment a repeat of the first?
Thank you for your question. The two experiments do not differ and were performed in direct chronological order on the HTP system. Yes, the second one was a repeat of the first. To obtain a number of four biological replicates per genotype and treatment, both experiments were performed.
- Relative humidity needs to be mentioned as this determines water use. Also, the 20 deg C minimum temperature seems high for most chickpea growing environments. What actual chickpea growing environment was the controlled environment facility relating to?
Thank you for this comment. For the settings, we have used previously published studies in the greenhouse as a guide (Atieno 2017, 2021). The relative humidity was described in line 144.
- L160,163. “Error! Reference source not found.”? And further throughout the text. Is this supposed to be indicating tables and figures?
Please excuse the errors in the formatting. Many of these errors relate to the supplementary material (SM). We have fully checked the numbering and cross references.
- The relevance of colour measurements needs to be further explained here.
The measurements were explained further in line 173-176.
- The symbols used should be defined.
L253. The relation between heritability and data quality needs further explanation, requiring elaboration of the equation at L219?
Thank you for your comment. We explained the equation in line 235.
- Figs 1-8 not referred to in text.
Please excuse the errors in the formatting. We have corrected these issues.
- Figs 5 and 6. Need to explain the meaning of the shapes around each data point.
Please excuse, we had overlooked this inscription before and now it is in line 336 and 349.
- L283 and later. What is SM?
Please excuse, SM means supplementary material. Abbreviation is explained in line 118.
- Fig. 1 repeated, should it be Fig. 6?
Yes, please excuse the wrong numbering and formatting.
- Figs 7 and 8. Indicate which points are identified as drought tolerant.
Thank you for this comment. We added the labels for the four superior genotypes for figure 7 and 8 (line 414 and 427).
Discussion
- Concentrates on evaluating use of HTP for chickpea and description of the overall (across genotypes) response to drought stress and recovery. However, very little space is given to discussion of genotypic differences, L611-617 only, which is surely the main objective of the study.
Thank you for your comment. We can understand this argument and have decided to omit the first part of the title.
“Engaging precision phenotyping to scrutinize vegetative drought tolerance and recovery in chickpea plant genetic resources”
is the title instead of:
“Searching for donor genotypes in chickpea: utilizing precision phenotyping to scrutinize vegetative drought tolerance throughout plant genetic resources”
In addition, line 9-12, 24-28, 108-114 and 612-618 were rewritten.
References
- Reference Nos. 14 and 15 are the same.
Please excuse the circumstances, we corrected the references.

Reviewer 2 Report
Journal: Plants (ISSN 2223-7747)
Manuscript ID: plants-2456699
Type: Article
Title: Searching for donor genotypes in chickpea: utilizing precision phenotyping to scrutinize vegetative drought tolerance throughout plant genetic resources
Climate change, food shortage, water scarcity, and population growth are some of the threatening challenges being faced in today’s world. Drought stress poses a constant challenge for agricultural crops and has been considered a severe constraint for global agricultural productivity; its intensity and severity are predicted to increase in the near future. Legumes demonstrate high sensitivity to drought stress, especially at vegetative and reproductive stages. They are mostly grown in the dry areas and are moderately drought tolerant, but severe DS leads to remarkable production losses. The most prominent effects of DS are reduced germination, stunted growth, serious damage to the photosynthetic apparatus, decrease in net photosynthesis, and a reduction in nutrient uptake. To curb the catastrophic effect of drought stress in legumes, it is imperative to understand its effects, mechanisms, and the agronomic and genetic basis of drought for sustainable management.
This topic is investigated in the literature, and there is a very few of reference published. However, this paper gives significant contribution to the current knowledge in related field. The data are sound and it deserves to be published, after major revisions.
Overall Recommendations: Major Revisions
Title
??? Title may be changed based on the findings of this study;
Abstract
??? Abstract should be re-written with an opening sentence pertaining to the generalized issue, what was done in the study, key findings and implications of the results should be clearly stated in the abstract.
Introduction
??? Authors completely failed to develop the hypothesis with reference to title and objective, in the introduction section. Why the present research and manuscript is planned and written??? Otherwise, it looks like a B.Sc. student’s assignment.
??? Kindly delete the sub-headings. Such as
1.1. Chickpea and drought (Page-2, Line-46)
1.2. Impact of drought on photosynthesis (Page-2, Line-70)
1.3. HTP and drought (Page-2, Line-78)
1.4. Objective (Page-3, Line-97)
Materials and methods
??? The text has many typing and grammatical errors, capitalization issues.
??? All proper nouns must be abbreviated. Abbreviations must be described completely at first mention with brackets. Don’t start a sentence with an abbreviation here???
??? The materials and methods section is very brief. Please add details for analytical methodologies to make it reproducible.
??? Quality assurance of data is mandatory!!! How many batch, repeats, chemical grade and for used instruments manufacturers’ user manual and instructions were strictly followed or not!!!
??? The methodology section warrants concise description followed by only one or two names of the reference manuals used for the analyses. Statistical methods should be briefly described and does not mingle with other technical obtained data.
Results
Data is sound one. It deserves to be published.
??? Elaborated one. But all speculations, no confirmation made through repeated experiment.
Tried well but inconsistent and no link with parameters, as well as no logical connection has been made with previous findings???
Discussion
??? The results and discussion should be written succinctly with support from relevant references only. Chickpea is the test crop in this study and the references used should be predominantly related to chickpea.
Conclusion
??? The conclusion section must not be the results section second window. Novelty of this research work is again questionable with reference to practical significance and economic feasibility must be worked and mentioned.
References
??? A few very old references have been used. These must be updated with recent research findings or removed.
??? Proper formatting is questionable. It must be according to MDPI Plants Journal.
??? References formatting are inconsistent. A few DOI missing??? Verify each reference from original source and cross check references in the text and reference section.
??? English style and language requires a profound revision. However, the readability of the manuscript needs to be improved, preferably carefully reviewing by a native English speaker???
Author Response
My answers are written in red. The attachted file is a revised version of the manuscript.
The linguistic improvements are highlighted in turquoise, required changes in pink and improvements, which I noticed in yellow.
Title
- ??? Title may be changed based on the findings of this study;
Thank you for your comment. We can understand this argument and have decided to omit the first part of the title.
“Engaging precision phenotyping to scrutinize vegetative drought tolerance and recovery in chickpea plant genetic resources”
is the title instead of:
“Searching for donor genotypes in chickpea: utilizing precision phenotyping to scrutinize vegetative drought tolerance throughout plant genetic resources”
In addition, line 108, 113 and 612 were rewritten.
Abstract
- ??? Abstract should be re-written with an opening sentence pertaining to the generalized issue, what was done in the study, key findings and implications of the results should be clearly stated in the abstract.
Thank you very much for this thought. We have rewritten the abstract accordingly (line 9 and 24).
Introduction
- ??? Authors completely failed to develop the hypothesis with reference to title and objective, in the introduction section. Why the present research and manuscript is planned and written??? Otherwise, it looks like a B.Sc. student’s assignment.
We can understand your point of view to some extent and have revised the lines 9, 24, 40, 42, 71, 108 and 113, accordingly.
- ??? Kindly delete the sub-headings.
Please excuse the inconvenience, we have now deleted the subheading
Materials and methods
- ??? The text has many typing and grammatical errors, capitalization issues.
Thank you for reading the text so carefully. After going through the text again, we asked a native speaker to correct the language. Please find these changes highlighted in turquoise.
- ??? All proper nouns must be abbreviated. Abbreviations must be described completely at first mention with brackets. Don’t start a sentence with an abbreviation here???
Thank you for this comment to improve the clarity of the text. We have looked through all the traits and repetitive terms and checked the abbreviations and explanation of the acronyms at the beginning.
- ??? The materials and methods section is very brief. Please add details for analytical methodologies to make it reproducible.
??? Quality assurance of data is mandatory!!! How many batch, repeats, chemical grade and for used instruments manufacturers’ user manual and instructions were strictly followed or not!!!
Thank you for this comment to describe the study more precisely and make it reproducible. Since these two questions go in the same direction from my point of view, I would like to explain this together. The plant material consists of 60 genotypes, which are described in line 117-130 and supplementary material (SM 1). Each measurement was done for one carrier and each carrier contained one plant, which was described in line 138-142. The used substrate and fertilizer were described in line 142 and 147-148. In each of the two experiments there were 2 biological replicates per genotype and treatment. This results in four biological replicates per genotype and treatment in two identical experiments. For the daily imaging and the image-derived traits, the software from the manufacturer LemnaTec was used as standard for the control and the Integrated Analysis Platform (IAP) was used for the image analysis, as described (line 132-134 and 170-184). The chlorophyll fluorescence measurements were carried out with a chlorophyll fluorescence camera from the manufacturer PSI (line 204-206). The analysis was carried out as standard with Plant Data Analyzer (line 206-207) and the protocol was described in line 213-223.
- ??? The methodology section warrants concise description followed by only one or two names of the reference manuals used for the analyses. Statistical methods should be briefly described and does not mingle with other technical obtained data.
The packages used for R studio were listed in line 227-230.
In addition, the subchapters have been completely reordered and renamed so that the technical obtained data is no longer mixed with the statistics (line 115-252).
Results
Data is sound one. It deserves to be published.
- ??? Elaborated one. But all speculations, no confirmation made through repeated experiment.
Tried well but inconsistent and no link with parameters, as well as no logical connection has been made with previous findings???
We have already presented two experiments here, i.e. we have already repeated the experiment. As high-throughput phenotyping is a fairly new field, there are not many published studies that accurately describe the traits. So far, there are only two studies with chickpea and HTP in a greenhouse pot experiment published (line 96). In follow-up experiments, we have included some metabolomics sampling compare our data with data from field experiments and further dissect the effects.
- Discussion
??? The results and discussion should be written succinctly with support from relevant references only. Chickpea is the test crop in this study and the references used should be predominantly related to chickpea.
Thank you for this comment. As mentioned before (question 9.), HTP is a relatively new topic and there is limited literature, especially for chickpeas (line 96). Nevertheless, we have added studies on chickpeas, i.e. line 45.
Conclusion
- ??? The conclusion section must not be the results section second window. Novelty of this research work is again questionable with reference to practical significance and economic feasibility must be worked and mentioned.
Thank you for this comment, we have revised the conclusion and tried to infer the focus and originality and relevance of the study (line 612).
References
- ??? A few very old references have been used. These must be updated with recent research findings or removed.
Thank you for the suggestion. We have looked at all references again, checked whether they are relevant or not.
- ??? Proper formatting is questionable. It must be according to MDPI Plants Journal.
??? References formatting are inconsistent. A few DOI missing??? Verify each reference from original source and cross check references in the text and reference section.
Thank you for the suggestion. The article has been completely adapted to the formatting requirements of MDPI Plants, including the listing of references.
Comments on the Quality of English Language
- ??? English style and language requires a profound revision. However, the readability of the manuscript needs to be improved, preferably carefully reviewing by a native English speaker???
We apologize for the inconvenience. The manuscript was read and revised by a native speaker. The changes are marked in turquoise.

Round 2
Reviewer 1 Report
An elegant, detailed study giving insight into the physiology of drought stress and recovery in chickpea. However, the relevance of the genotypic differences found to improving drought resistance/tolerance of chickpea needs to be further clarified.
Author response numbering
1. It should be mentioned in the article that “field experiments with the same genotypes are being conducted by our collaboration partners in Italy” and the steps involved in integrating these studies to ultimately produce more drought resistant genotypes.
2. It needs to be acknowledged in the Discussion that above-ground physiology is just one component of chickpea response to drought stress. Rooting ability of chickpea has been shown to be a major determinant of drought response.
3. Yes, nodulation is mentioned in the Introduction but nodulation status of plants in the study is not mentioned, nor discussed as to what effect it may have had on the obtained results.
4. Yes, this study does give some further insight into the physiological differences in drought response of kabuli and desi types and superior genotypes are indicated in Figs 7 and 8.
5. Criteria for assessing drought response remain unclear. Is it ability to maintain EB under drought stress or relative reduction under stress compared to the control; or something else. Of course, one may expect to find genotypic difference for all traits measured under all growth conditions but the question for plant improvement is whether these differences are large enough to likely improve field performance under drought. This paper needs to address that question.
6. Although this study elucidates particular physiological responses to drought stress and recovery, it needs to better lay out the pathway of carrying through this information to improving drought response in chickpea; whether through use of QTLs or conventional breeding. The pathway should be clarified in the Discussion or Conclusion.
7. Needs to be made clearer in the text that the 2nd experiment was a repeat of the 1st, to give a total of four replications.
8. “Yes, the second one was a repeat of the first. To obtain a number of four biological replicates per genotype and treatment, both experiments were performed.” OK, but this needs to be included in the text.
9-17. Now explained.
18. The revised title helps but it remains necessary to lay out the pathway from this HTP data to practical improvement of drought stress in chickpea.
Additional comments on revised manuscript
Abstract
L10-13. Long and complex sentence likely to discourage a potential reader. Break to at least two sentences.
L25-26. “wide phenotypic diversity”? Disagree, Figs. 3 and 4 indicate low diversity.
Discussion
L512-6. Unclear. Tall plants are an advantage for mechanical harvesting, which would likely outweigh the risk of weather damage. What does “storage propensity” mean?
Entirely understandable but could be improved by minor English language editing
Author Response
Dear Reviewer,
Thank you for giving us the opportunity to submit a revised draft of the manuscript. We appreciate the time and effort that you dedicated to providing feedback on our manuscript and are grateful for the insightful comments on and valuable improvements to our paper.
We have incorporated most of the suggestions made by you.
The linguistic improvements are highlighted in turquoise, required changes in pink and improvements, which I noticed in yellow.
Author response numbering
- A It should be mentioned in the article that “field experiments with the same genotypes are being conducted by our collaboration partners in Italy”
Thank you for pointing this out, we added this information in line 127.
- B and the steps involved in integrating these studies to ultimately produce more drought resistant genotypes.
Thank you very much for this suggestion. We have rewritten the conclusion to describe the integration of this study into the targeted improvement crop, please see line 616-630.
- It needs to be acknowledged in the Discussion that above-ground physiology is just one component of chickpea response to drought stress. Rooting ability of chickpea has been shown to be a major determinant of drought response.
Thank you for this suggestion. We mentioned the rooting ability in line 81 and 623.
- Yes, nodulation is mentioned in the Introduction but nodulation status of plants in the study is not mentioned, nor discussed as to what effect it may have had on the obtained results.
Thank you, we have added the suggested content in line 152-153 and in line 283-285.
- Yes, this study does give some further insight into the physiological differences in drought response of kabuli and desi types and superior genotypes are indicated in Figs 7 and 8.
Thank you for paying attention to this improvement!
- Criteria for assessing drought response remain unclear. Is it ability to maintain EB under drought stress or relative reduction under stress compared to the control; or something else. Of course, one may expect to find genotypic difference for all traits measured under all growth conditions but the question for plant improvement is whether these differences are large enough to likely improve field performance under drought. This paper needs to address that question.
Thank you for pointing this out, in line 10 and 81-83 we go further into detail.
With regard to the differences please read comment 21 below and, in the conclusion, we address the crop improvement.
- Although this study elucidates particular physiological responses to drought stress and recovery, it needs to better lay out the pathway of carrying through this information to improving drought response in chickpea; whether through use of QTLs or conventional breeding. The pathway should be clarified in the Discussion or Conclusion.
Thank you very much for this suggestion. As indicated for the comment 1B, we have rewritten the conclusion to describe the integration of this study into the targeted improvement crop, please see line 616-630.
- Needs to be made clearer in the text that the 2ndexperiment was a repeat of the 1st, to give a total of four replications.
We think this is an excellent suggestion and added this information in line 143.
- “Yes, the second one was a repeat of the first. To obtain a number of four biological replicates per genotype and treatment, both experiments were performed.” OK, but this needs to be included in the text.
As mentioned in the answer for the comment 7., we added this information in line 143. Thank you.
9-17. Now explained.
Thank you!
- The revised title helps but it remains necessary to lay out the pathway from this HTP data to practical improvement of drought stress in chickpea.
Thank you for this comment. As answered for comment 1B, 5 and 6 we have rewritten the conclusion to address this point.
Additional comments on revised manuscript
Abstract
- L10-13. Long and complex sentence likely to discourage a potential reader. Break to at least two sentences.
Following your suggestion, we have changed these lines. Thank you, for bringing this to our attention.
- L25-26. “wide phenotypic diversity”? Disagree, Figs. 3 and 4 indicate low diversity.
Thank you for this pointing this out. Please note that the shading around the line for the control and drought stress treatment indicate the confidence interval. This is clear from the description of the Figures. We show the confidence intervals to show the significant difference between the treatments. The variation was described in the result section 3.1, in the Supplementary Material (SM) 7 in the descriptive statistics and in the SM 11 with the coefficient of variation. Based on this data, we disagree with the statement of low diversity and clearly see high diversity.
Discussion
- L512-6. Unclear. Tall plants are an advantage for mechanical harvesting, which would likely outweigh the risk of weather damage. What does “storage propensity” mean?
Thank you very much for this comment. We have reviewed and corrected this paragraph.
Reviewer 2 Report
The manuscript in present form has been sufficiently improved to warrant publication in Plants by the authors.
Author Response
Dear Reviewer,
Thank you for giving us the opportunity to submit a revised draft of the manuscript. We appreciate the time and effort that you dedicated to providing feedback on our manuscript and are grateful for the insightful comments on and valuable improvements to our paper.
With best wishes!
Round 3
Reviewer 1 Report
The authors have made satisfactory revisions to address this reviewer's previous comments.